# Cascade Forest-Based Model for Prediction of RNA Velocity

**DOI:** 10.3390/molecules27227873

**Published:** 2022-11-15

**Authors:** Zhiliang Zeng, Shouwei Zhao, Yu Peng, Xiang Hu, Zhixiang Yin

**Affiliations:** School of Mathematics, Physics and Statistics, Shanghai University of Engineering Science, Shanghai 201620, China

**Keywords:** RNA velocity, scRNA-seq, cascade forest, ensemble classifier

## Abstract

In recent years, single-cell RNA sequencing technology (scRNA-seq) has developed rapidly and has been widely used in biological and medical research, such as in expression heterogeneity and transcriptome dynamics of single cells. The investigation of RNA velocity is a new topic in the study of cellular dynamics using single-cell RNA sequencing data. It can recover directional dynamic information from single-cell transcriptomics by linking measurements to the underlying dynamics of gene expression. Predicting the RNA velocity vector of each cell based on its gene expression data and formulating RNA velocity prediction as a classification problem is a new research direction. In this paper, we develop a cascade forest model to predict RNA velocity. Compared with other popular ensemble classifiers, such as XGBoost, RandomForest, LightGBM, NGBoost, and TabNet, it performs better in predicting RNA velocity. This paper provides guidance for researchers in selecting and applying appropriate classification tools in their analytical work and suggests some possible directions for future improvement of classification tools.

## 1. Introduction

With the rapid development and innovation of single-cell RNA sequencing (scRNA-seq) technology and related bioinformatics methods [1,2,3,4,5,6,7,8], high-throughput single-cell data are emerging in large quantities, and “scRNA-seq technology” has become an important tool for molecular biology research. Compared with traditional cell-based RNA-seq, scRNA-seq can better reflect the molecular biological processes within a specific cell population. Currently, the use of single-cell RNA sequencing (scRNA-seq) data for cell trajectory inference is one of the pressing research issues [9,10,11,12,13,14,15]. Many researchers have proposed many algorithms for cell trajectory inference, for example, TSCAN [16] constructs a minimum spanning tree (MST) based on the center of mass of a cluster of cells and then infers the pseudo-temporal order of cells. There is also a class of algorithms to infer trajectories based on the graph structure. For instance, Wanderlust [17] and SLICER [18] construct intercellular graphs based on KNN graphs, and SoptSC [19] uses intercellular similarity matrices for inferring trajectories.

La Manno et al. [20] introduce the concept of RNA velocity. This concept develops new ways to study cellular dynamics by linking data to underlying molecular dynamics. Therefore, the predictive models of cellular dynamics can be achieved. RNA velocity is an indicator of transcript dynamics that predicts future changes in cellular state, and it allows for reliable estimation of relative temporal derivatives of gene expression state for studies of cell differentiation, lineage development, and dynamic changes in cellular components in the tumor microenvironment. RNA velocity recovers targeted information by distinguishing between newly transcribed pre mRNA (unspliced) and mature mRNA (spliced), the latter of which can be detected from the presence of introns in standard single-cell RNA-seq protocols. The change in mRNA abundance is referred to as RNA velocity. La Manno et al. [20] propose the original steady-state model algorithm, named velocyto, for RNA velocity assumes that the transcriptional phase lasts long enough to reach steady-state equilibrium and that equilibrium mRNA levels can be approximated by simplifying linear regression with co-splicing rates. Volker Bergen et al. [21] propose a model called scVelo algorithm, which includes a stochastic, kinetic model and steady-state model.

Wang et al. [22] take a bold stab at RNA velocity research by using RNA velocity prediction as a supervised learning problem for classification, dividing the cell state space into segments of equal size by direction as classes. The estimated RNA velocity vector is treated as ground truth, which we counted as a true positive if the predicted direction is located in the same segment as the original target direction, and so on. In this paper, we study the supervised learning problem based on RNA velocity prediction classification using internally generated mixed cell lineage datasets and a number of complex public datasets, which are well annotated with cell-type labels. We evaluate six integrated classifiers (XGBoost [23], RandomForest [24], LightGBM [25], NGBoost [26], TabNet [27], Cascade Forest [28]) in the problem of RNA velocity prediction. This can make biologically meaningful predictions.

The paper is organized as followed. The second section introduces the concept and theory of RNA velocity estimation. The cascade forest structure model and the evaluation indicators to be used in subsequent experiments are introduced in this chapter. Finally, the information of the relevant data set is briefly described. In the third section, experimental results are analyzed, such as preprocessing the data set, evaluating the experimental results of the data set, using accuracy, the kappa coefficient, and evaluating the benchmark performance of the classifier. Finally, the stability of the performance of the cascade forest model is further analyzed by parameter comparison. In the fourth and fifth parts, we review the classification problem of speed prediction and suggest the future direction of classification tools. We believe that the research on RNA velocity prediction will guide the classification of cell velocity prediction and classification tools for scRNA-seq datasets in various use cases.

## 2. Result

### 2.1. Data Set

A comprehensive and systematic evaluation of the RNA velocity prediction classification base classifier requires scRNA-seq datasets with well-annotated cell labels, as most of the evaluation metrics rely on the underlying factual label set for their calculation. Therefore, only scRNA-seq datasets with highly plausible cell-type labels are used in this paper. In the numerical experiments in this paper, datasets with different feature dimensions and different numbers of classes were carefully selected to better characterize the performance of the integrated algorithm in the RNA velocity prediction problem.

Gastrulation_e75: A subset of mouse gastrulation is E7.5, in embryonic day 7.5 (E7.5) mutants, revealing a molecular map of mouse gastrulation and early organogenesis, including 7202 from mouse embryos, and a transcriptional profile of 53,801 single cells [29].

Bonemarrow: A human bone marrow cell dataset which consists of hematopoietic cells, bone marrow adipose tissue, and supporting stromal cells, and includes transcript levels of 14,319 genes in 5780 cells [30].

Pancreas: A dataset from the embryonic pancreas of NVF homozygous mice, with transcript levels of 27,998 genes from 3696 pancreatic epithelial and Ngn3-Venus fusion cells [30].

Dentategyrus: A mouse hippocampal dentate gyrus neurogenesis dataset. Comprising RNA-seq data of 13,913 genes and 2930 cells from multiple lineages, the dominant structure is the granulosa cell lineage in which neuroblasts develop into granulosa cells. The remaining populations form distinct cell types that are fully differentiated (e.g., Cajal-Recius cells) or form sub-lineages [31].

As shown in Figure 1, we plotted the splicing ratios of each dataset, and 10–25% of the unspliced molecules in different data using single-cell datasets typically contain intronic sequences. 

### 2.2. Data Preprocessing

As with traditional scRNA-seq analysis, measurements require preprocessing. First, genes were filtered by expression level or their occurrence in cells, retaining genes with sufficient counts, selecting the top 2000 highly variable genes for datasets with less than 3000 cells and then for data with less than 6000 cells and greater than 3000 cells. The top 2500 highly variable genes were selected for the set and, finally, the top 3000 highly variable genes were selected for cells with more than 6000 cells, as shown in Table 1, which provides the specific description of each data set. 

For highly variable genes, the expression levels of these genes vary greatly between different cells (highly expressed in some cells and low in others). Screening out highly variable genes can reduce the noise of the dataset to a certain extent. After the first filter, normalization was performed to handle differences in sequencing depth between individual cells, followed by a log transformation, and highly variable genes were identified and extracted, while all remaining genes were discarded. Finally, according to the nearest neighbor graph, the first and second moments were calculated. There are three methods for velocity estimation in scVelo: the steady-state model, stochastic model, and dynamic model. 

In this article, we evaluated the classification under the steady-state model and the dynamic model, respectively. From the velocity estimates, we could obtain a multidimensional RNA velocity vector $V=\left\{v_{1},v_{2},\cdots v_{n}\right\}$ for each transcriptional state of a single cell. In order to extract relevant signals and infer RNA velocity, the feature selection is performed according to the gene ranking of ScVelo and the first k gene in each cluster is selected as the model feature and, finally, assigned d class labels by an equal division of a 2D circular plane. In the RNA velocity prediction classification problem, in order to better evaluate the effect of each classifier, we set the parameters by default k=20,d=4; that is, we selected the top 20 genes in each cluster to divide the RNA speed prediction into four directions.

### 2.3. Performance Evaluation

For classification performance evaluation in the steady-state mode first, we trained and tested models on four single-cell RNA-seq datasets, dividing the dataset into training and test datasets in an 8:2 ratio, and then matched the training set to these classifier models to extract and learn hidden patterns. Subsequently, the learning of the test dataset was evaluated against accuracy, F1_scores, and kappa scores, and the prediction results were retrieved. To address the data imbalance problem in this paper, an oversampling method SMOTETomek [32] was invoked, which can largely reduce the loss of information and improve the performance of the model. Figure 2 shows the classification accuracy, macro_F1, and Kappa coefficient for all tools in the steady-state mode for the four test cases. A comparative study of all the specified integrated classifiers for the specified metrics shows that we can find that, among datasets with different clipping ratios, the cascade forest model achieves relatively good results, followed by XGBoost and Random Forest, which also perform well in the steady-state mode.

We then ran the dynamics model to learn the full transcriptional dynamics of the splicing dynamics, which was solved in a likelihood-based expectation maximization framework by iteratively estimating the parameters of the response rate and latent cell-specific variables (i.e., transcriptional state and intracellular latency time), and it can be seen in Figure 3 that the cascade forest model continues to perform better than other classifiers in the dynamics model, with good robustness and accuracy in all model. The accuracy, F1_score, and kappa are still the highest and have improved significantly, reaching 0.937 in the dentategruys dataset and 0.916 in F1_score and kappa coefficient, respectively. TabNet’s performance, in terms of accuracy and F1_score and kappa coefficients, in the dynamics model was average, and both classifiers seemed to be less sensitive to speed prediction classification in different model.

### 2.4. Comparison with Existing RNA Velocity Prediction Classification Methods

We compared the stacking model classifier proposed by Wang et al. [22]. We further investigated the effects of hyper-parameter k, which is the number of top genes, and parameter d, which is the number of categories, on cascade forest models and stacked models. In the feature selection part of the parameter k control, Figure 4 shows that when d is set to four, we discover that the accuracy of both methods can increase from the increase in k, which may be that with the increase in features, it contains more useful RNA information, which can make the RNA velocity prediction classification more accurate, and also shows that the accuracy of the cascade forest model in the four datasets is significantly better than that of the stacking model.

With the previous experimental results, we set k to 20, although we can increase k for better performance. Figure 5 shows that, as we continue to increase d, the task becomes more difficult in the stacking model, so the accuracy score decays and fluctuates greatly while the cascade forest still performs well and has good robustness relative to the stacking model. 

## 3. Discussion

This paper introduces a novel machine-learning algorithm cascade forest model for RNA velocity prediction and classification tasks. The cascade forest model is a decision tree ensemble approach that has fewer hyperparameters than deep neural networks and preserves the tree model’s interpretability. Its model complexity can be determined automatically in a data-dependent manner, making cascade forest models work well even on small-scale data. This paper aims to effectively predict RNA velocity using cascade forest models to prove that machine learning (ML) algorithms based on ensemble strategies are effective in predicting RNA velocities. In this paper, experiments were performed on four scRNA-seq datasets with different complexities and splicing ratios. The cascade forest model was comprehensively evaluated with five base classifiers: XGBoost, RandomForest, LightGBM, NGBoost, and TabNet. The experimental results show that the kappa coefficient, accuracy, and F1_score performance of the cascade forest model are significantly better than XGBoost, RandomForest, LightGBM, NGBoost, and TabNet, which proves that the model substantially improves RNA velocity prediction and classification.

In addition, this paper compares the cascade forest model with the stacking model proposed by Wang et al. for RNA velocity prediction and classification. Through parametric analysis, it is found that cascade forests have better stability and stronger robustness. Especially for the change of parameter d, the cascade forest model prediction accuracy will not fluctuate as much as the stacked model, and even the performance will show a significant downward trend. As parameter k increases, the number of features increases, and the cascade forest model can predict RNA velocity more accurately, showing better performance than the stacking model.

In this study, we extensively analyze RNA velocity prediction classification problems. Although the first application of existing RNA velocity models has shown promising results [33], the selection of characteristic genes and the screening of highly variable genes have been found in experimental tests to have different degrees of influence on speed prediction. Methods for assessing gene selection bias, joint models for better potential spatial representation, and factor models for unraveling compositional effects will be further studied, which will be necessary for future work.

Most classification tools accurately predict the results in RNA velocity prediction on four datasets with different splicing ratios. However, this result is based on an ensemble learning framework to balance different sample feature ratios to arrive at different baseline models, but it is still an empirical approach. When the data distribution is incomplete and unbalanced, the prediction results still have a more significant impact. In experimental tests on the Bonemarrow dataset, we found that the accuracy of cascade forest models decreased by more than 20% when the data were unbalanced. Therefore, the interpretability of the model will be a problem that needs to be solved in the future.

In conclusion, cascade forest models provide a new prediction-based method for studying the mechanisms of cell differentiation that can be applied to help attribute state spaces not yet covered by scRNA-seq data. In future work, we can use this interpolated direction information to conduct more in-depth research on trajectory inference. For example, differential geometry is used to extract potential adjustments for estimating the curvature of the differentiation landscape in metabolic labeling experiments [34]. Therefore, the prediction of RNA velocity can more intuitively understand the trend of cell dynamics, which has specific reference significance for biologists’ research.

## 4. Materials and Methods

### 4.1. RNA Velocity Estimation

The goal is to predict the RNA velocity vector for each cell based on its gene expression data, shown in 2D space, and to formulate this as a classification problem. For this problem, we first need to obtain a count matrix of unspliced and spliced mRNAs, which can be obtained using methods such as Velocyto [20], loompy/kallisto [35], or alevin [36], to obtain read annotations. After obtaining the count matrices, a simple kinetic model of the sheared expression can be built:(1)$$\frac{dU\left(t\right)}{dt}=\alpha \left(t\right)-\beta \left(t\right)U\left(t\right)$$
(2)$$\frac{dS\left(t\right)}{dt}=\beta \left(t\right)U\left(t\right)-\gamma \left(t\right)S\left(t\right)$$
where U(t) denotes the number of unspliced mRNA molecules, S(t) denotes the number of mRNA molecules after splicing, α denotes the transcription rate, β denotes the splicing rate from unspliced to spliced, and γ denotes the degradation rate of the mRNA product after splicing.

### 4.2. Steady-State Model

The steady-state model is one of the main models of RNA velocity. This model has two fundamental assumptions: (i) at the gene level, all full splicing dynamics with transcriptional induction, repression, and steady-state mRNA levels are captured; (ii) at the cellular level, all genes have a common splicing rate and gene expression follows steady state. At that time, the steady-state ratio is obtained, $\frac{dS\left(t\right)}{dt}=0\tilde{\gamma }=\frac{\gamma }{\beta }=\frac{u^{T}s}{|\left|s\right|{|}^{2}}$, where ||·|| represents the Euclidean distance.

At this time, the cell *i* velocity is estimated as the deviation of the ratio of spliced to unspliced molecules fitted to γ, which is:(3)$$v_{i}=u_{i}-\tilde{\gamma }s_{i}$$

### 4.3. Dynamic Model

In contrast, dynamic models directly solve for the complete transcriptional kinetics of each gene, rather than making transcriptome-wide assumptions. Instead of trying to fit the data into a regression model, it estimates the parameters using an Expectation Maximization algorithm (EM) [37] that uses the maximum likelihood to iteratively approximate α, β, and γ, and learns the spliced/unspliced trajectories of a given gene. This assigns the following likelihood function to each gene:(4)$$ℒ(\theta )=\frac{1}{\sqrt{2\pi \sigma }}\exp(-\frac{1}{2n}\sum_{i}^{n}\frac{||x_{i}^{obs}-x_{t_{i}}(\theta ){||}^{2}}{\sigma^{2}})$$
where $x_{i}^{obs}=\left(u_{i}^{obs},s_{i}^{obs}\right)\;$ represents the unspliced and spliced mRNA molecules of specific genes in the observed cell i, respectively. $x_{t_{i}}$ denotes the unspliced/spliced molecule i at time t based on the inferred parameter set θ=(α, β, γ).

### 4.4. Performance Metrics

In this paper, three predefined metrics are used to evaluate the performance of the five integrated classifiers: accuracy [38], macro_F1 [38], and kappa coefficient [39]. The above metrics can be considered as different combinations from the confusion matrix. Accuracy is a metric used to evaluate classification models and represents the proportion of the total number of correct predictions. The macro_F1 is a variant of the F1-Score in multiclassification evaluation, is a parameter calculated as a harmonic mean of accuracy and recall, and can combine the two metrics well. This paper also considers the Kappa coefficient, an important metric for assessing the accuracy of a multiclassification model.
(5)$$Accuracy=\frac{n_{correct\;}}{n_{total\;}}=\frac{TP+TN}{TP+FP+TN+FN}$$
(6)$$P_{i}=\frac{TP_{i}}{TP_{i}+FP_{i}}$$
(7)$$P_{macro}=\frac{\sum_{i=1}^{L}P_{i}}{\left|L\right|}$$
(8)$$R_{i}=\frac{TP_{i}}{TP_{i}+FN_{i}}$$
(9)$$R_{macro}=\frac{\sum_{i=1}^{L}R_{i}}{\left|L\right|}$$
(10)$$Macro\_F1=\frac{2P_{macro}R_{macro}}{P_{macro}+R_{macro}}$$
(11)$$Kappa=\frac{p_{0}-p_{e}}{1-p_{e}}$$
where *TP* (true positive) is the number of positive samples correctly identified. *FP* (false positive) is the number of negative samples identified as positive. *TN* (true negative) is the number of negative samples correctly identified. *FN* (false negative) is the number of positive samples missed. $p_{0}$ indicates the overall classification accuracy, $p_{e}$ indicates chance consistency, C is the total number of categories, and $T_{i}$ is the number of samples correctly classified for each category. Suppose that the real number of samples for each class is $a_{1},a_{2},\ldots ,a_{c}$ and the number of samples for each class predicted is $b_{1},b_{2},\ldots ,b_{c}$ and the total number of samples is n.

### 4.5. Classification Tool

In this study, we use several popular integrated enhancement algorithms, such as LightGBM [25], XGBoost [26], Random Forest [27], NGBoost [28], and TabNet [35], to evaluate the performance of each classifier in comparison with the cascade forest structure model [36].

After evaluating the five popular methods, we used four of the classifiers, as applied to the cascade forest structure, to predict with good robustness and accuracy. As shown in Figure 6, each level of the cascade forest structure consists of LightGBM, XGBoost, Random Forest, and NGBoost. These different types of integration methods are further integrated to increase diversity, which is crucial from an integration learning perspective. Each classifier will generate a class vector, which is generated by five cross-validations to avoid overfitting. The class vector is then concatenated with the original feature vector and received as input by the next level of the cascade. If the performance of the entire cascade does not improve significantly on the validation set, the propagation of the levels is terminated and therefore the complexity of the model can be determined automatically.

### 4.6. Availability of Data and Materials

scRNA-seq datasets are all public datasets and can be accessed through https://scvelo.org (accessed on 21 June 2022) direct interview. The hippocampal dentate gyrus neurogenesis dataset at P12 and P35 is available from the Gene Expression Omnibus (GEO) with accession number GSE95753. Atlas of mouse gastrulation is available under accession number GSE87038. Mouse pancreas data is also available from NCBI GEO, accession ID GSE132188. Human bone marrow data is available through the Human Cell Atlas data portal.

## 5. Conclusions

In summary, the supervised learning problem of RNA velocity prediction problem as classification is an entirely new study, and researchers still face many challenges. This study’s comprehensive evaluation was made using accuracy, F1_score, and kappa coefficients. We have demonstrated that the cascade forest model can work well in RNA velocity prediction classification problems, and its performance is better than XGBoost, RandomForest, LightGBM, NGBoost, TabNet, and stacking models, which are currently popular classification algorithms. Compared with the above classification algorithms, the cascade forest model is more stable and robust. Therefore, the cascade forest model can be better applied to any scRNA-seq data that can estimate RNA velocity. It guides researchers in selecting and applying appropriate classification tools in their analytical work and provides some possible directions for future improvements to classification tools.

Although the problem of modeling dynamics in scRNA-seq using RNA velocity is tricky, the test results in this paper suggest that it is feasible to incorporate classification into single-cell velocity predictive analysis workflows. Cascade forest models allow us to predict and infer the future expression of individual cells more accurately. In current applications, we can use it to analyze influencing genes in trajectory branching events. There are many other applications of this analysis that are worth exploring. We demonstrate that the cascade forest model substantially improves over previously proposed RNA velocity prediction methods on relevant real-world datasets from human and mouse developmental brains.

## Figures and Tables

**Figure 1 molecules-27-07873-f001:**
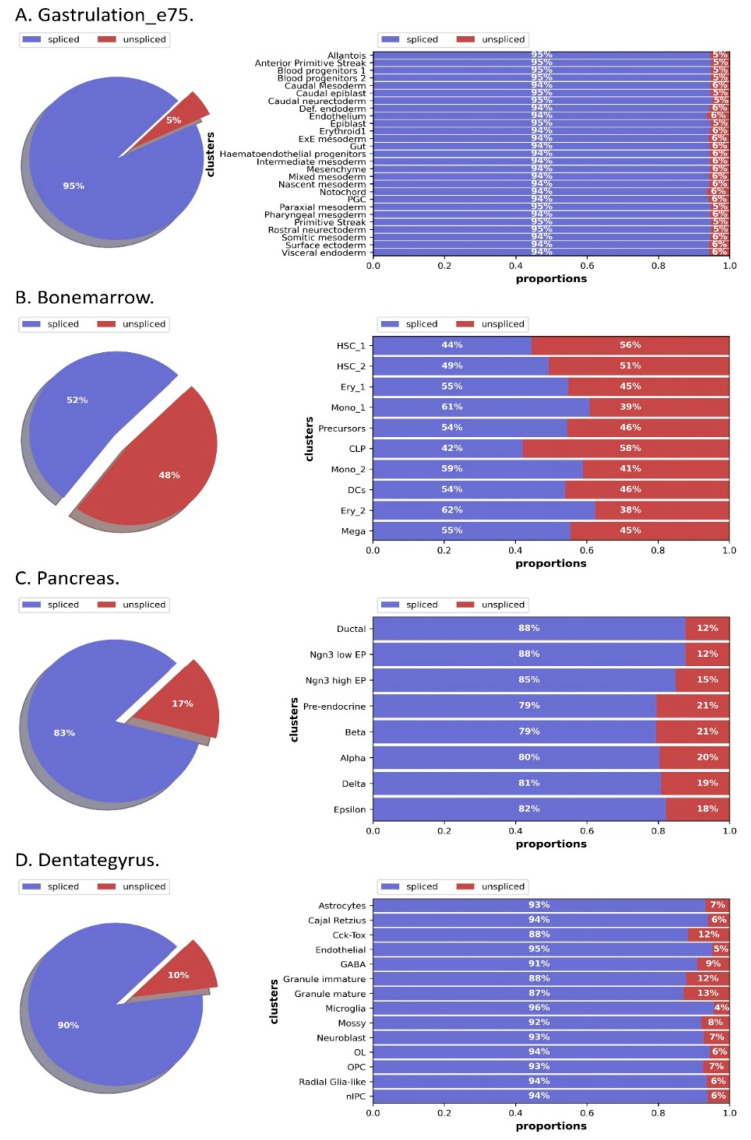
The distribution of spliced to unspliced proportions of cell types in different datasets.

**Figure 2 molecules-27-07873-f002:**
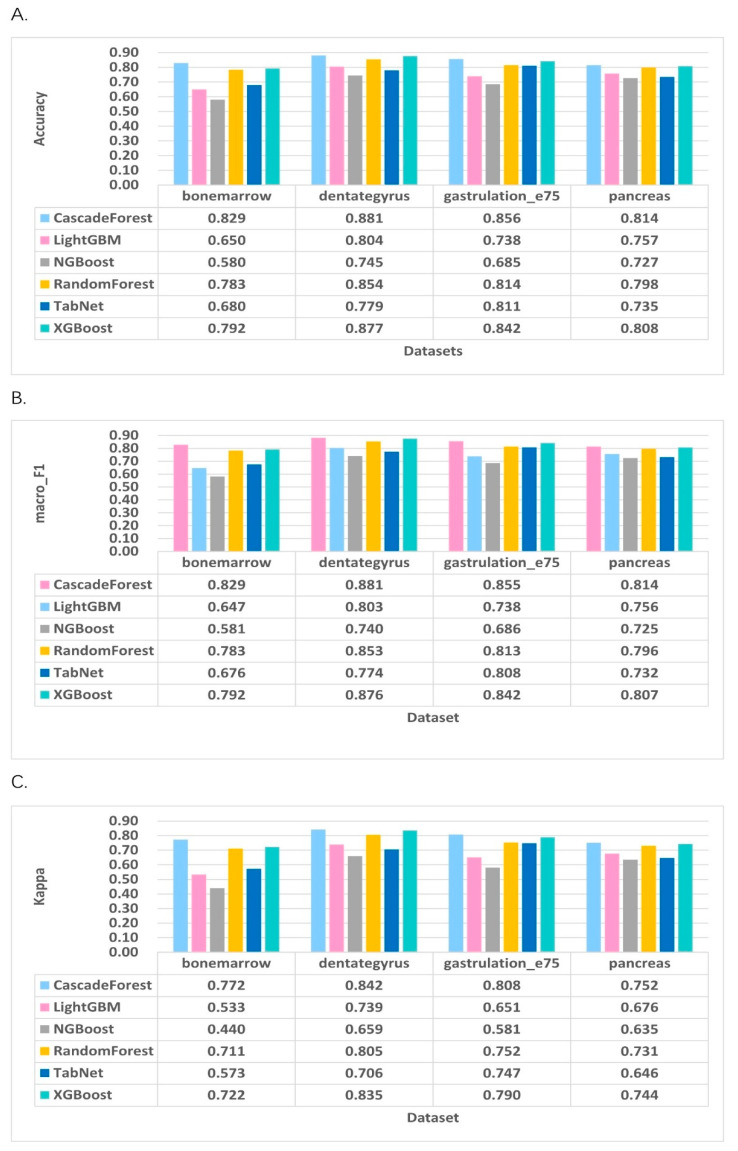
Evaluation of classification performance indicators for steady-state model RNA velocity prediction. (**A**) Accuracy of the classifiers. (**B**) Macro_F1 of the classifiers. (**C**) Kappa coefficients of the classifiers.

**Figure 3 molecules-27-07873-f003:**
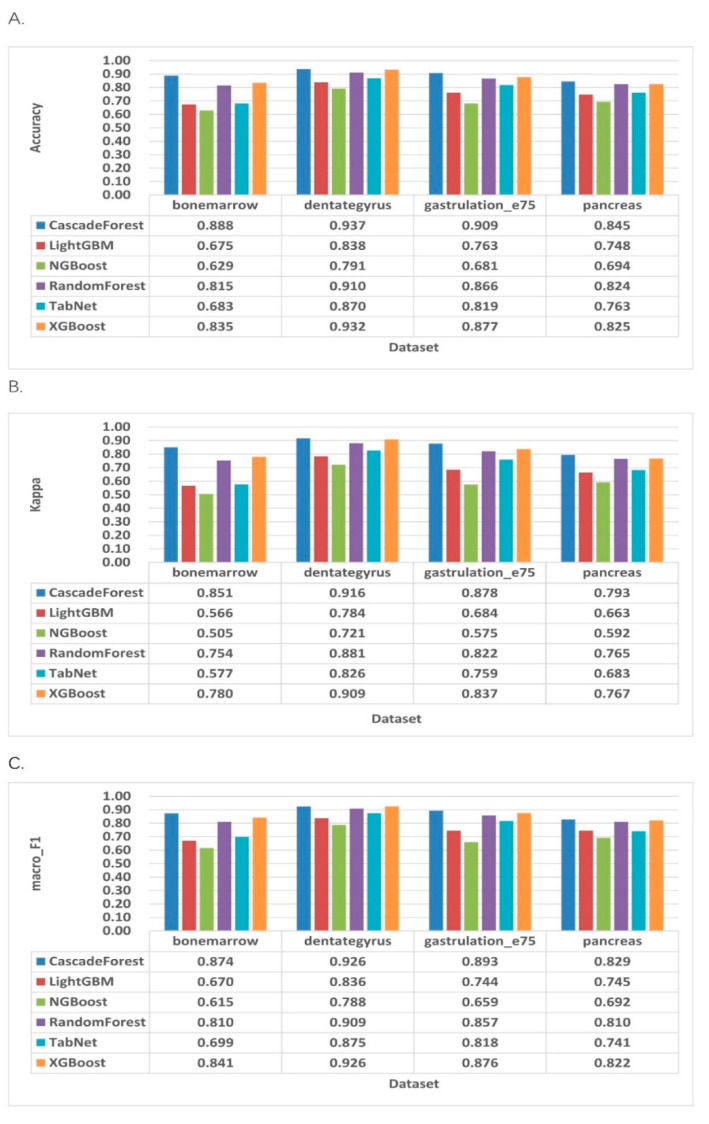
Evaluation of classification performance indicators for dynamic model RNA velocity prediction. (**A**) Accuracy of the classifiers. (**B**) Macro_F1 of the classifiers. (**C**) Kappa coefficients of the classifiers.

**Figure 4 molecules-27-07873-f004:**
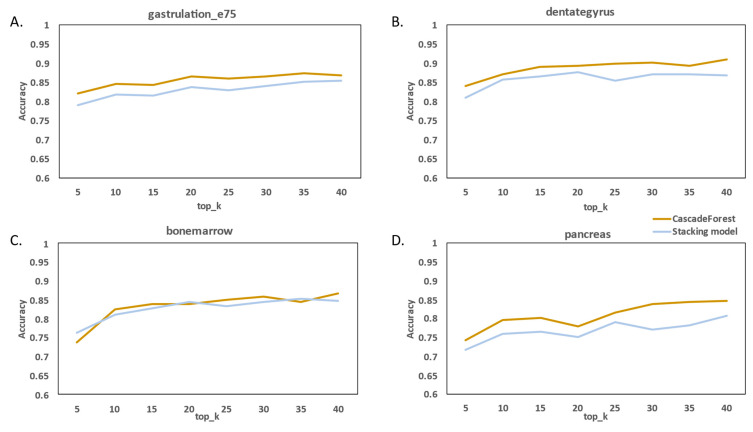
The effects of hyper-parameter k. (**A**–**D**) Performance of cascade forest models and stacking models under different datasets and different parameters k.

**Figure 5 molecules-27-07873-f005:**
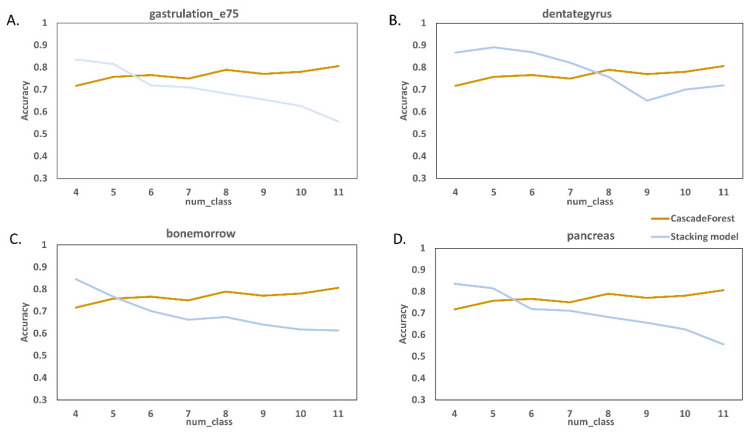
The effects of hyper-parameter d. (**A**–**D**) Performance of cascade forest models and stacking models under different datasets and different parameters d.

**Figure 6 molecules-27-07873-f006:**
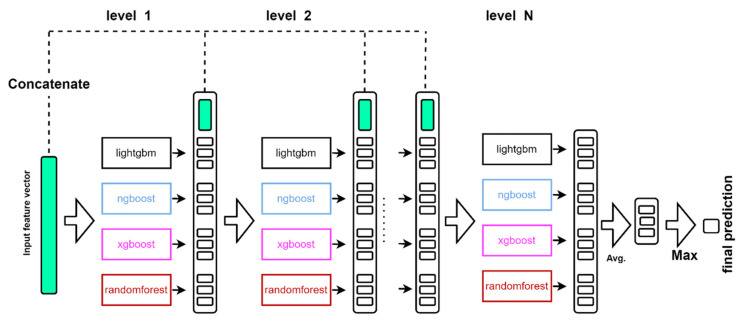
Diagrammatic representation of the cascade forest structure.

**Table 1 molecules-27-07873-t001:** Description of cells and genes in each dataset.

Datasets	Cell Number	Gene Number	Highly Variable Genes	Feature Numbers (*k* = 20)
gastrulation_e75	7202	53,801	3000	291
bonemarrow	5780	14,319	2500	141
pancreas	3696	27,998	2500	143
dantategrus	2930	13,913	2000	151

## Data Availability

Data are contained within the article.

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
