# Peer review of "Cascade Forest-Based Model for Prediction of RNA Velocity"

_molecules, 2022, doi:10.3390/molecules27227873_

Round 1
Reviewer 1 Report
Comments on molecules-2007456
I have reviewed the manuscript entitled: Cascade Forest-based for Prediction of RNA Velocity. Predicting the RNA velocity vector for each cell based on the gene expression data for each cell and formulating the RNA velocity prediction as a classification problem is a new direction of research. This paper is a further extension of the RNA velocity prediction classification problem to provide a more robust classifier, offering some possible directions for future improvements in classification tools. It is also the first time that a cascade forest model has been applied to the RNA velocity prediction problem with good results.
However, there are still shortcomings in this paper that need to be improved.
1. In the performance evaluation section of the results section, something needs to be added to further justify the results of the experiment in order to better demonstrate the superior performance of the cascade forest model.
2. the formatting of the images, the colour scheme and the content of the notes do not seem to be good enough and may need further adjustment.
3. the references need to be augmented with some relevant research literature that was previously missed.
4. the formatting of the formulae in the text is not particularly standardized and needs to be revised.
Author Response
Dear Reviewer:
Thank you for your comments concerning our manuscript entitled “Cascade Forest-based for Prediction of RNA Velocity.” Those comments are all valuable and very helpful for revising and improving our paper, as well as the essential guiding significance to our research. We have studied the comments carefully and made a correction that we hope meets with approval. Revised portions are marked in red on the paper. The primary corrections in the paper and the responses to the reviewer’s comments are as follows:
Responds to the reviewer’s comments:
Point1: In the performance evaluation section of the results section, something needs to be added to justify the results of the experiment further in order better to demonstrate the superior performance of the cascade forest model
Response1: Considering the Reviewer’s suggestion, We have enhanced the explanatory nature of the article. The original parametric analysis section has been deleted at the end of the results section. We add a new subsection, mainly to compare the cascade forest model with the existing methods for a parameter comparison through horizontal and vertical comparison, more comprehensively showing the excellence of cascade forest performance.
Point2: the formatting of the images, the color scheme, and the content of the notes do not seem to be good enough and may need further adjustment.
Response2: We have made corrections according to the Reviewer’s comments.
In the above changes, we have revised and added two more straightforward comparison charts of the experimental results and modified the caption in the image in the previous document.
Point3: the references need to be augmented with some relevant research literature that was previously missed.
Response3: We are very sorry for our negligence of references. We have added several more research pieces of literature to the introduction.
Point4: the references need to be augmented with some relevant research literature that was previously missed.
Response4: We are very sorry for our incorrect writing. The formulas in the previous manuscript were not center-aligned but have now been corrected, and all formulas are in the same font size as they are.
Special thanks to you for your good comments. We tried our best to improve the manuscript and made some changes. These changes will not influence the content and framework of the paper. And here, we did not list the changes but marked them in red in the revised paper. We appreciate the Reviewers’ warm work earnestly and hope the correction will be approved. Once again, thank you very much for your comments and suggestions.
Reviewer 2 Report
The paper makes a contribution to the field of medicine. This version of the manuscript has several weaknesses in the organizational part, which made the paper hard for the reader. My suggestions to the authors are listed below:
I need clarification on why the authors set the material and method as section 4.
My recommendation to the authors is to reorganize the paper in the following way:
1. Introduction (Section 1) (In this section, add the sentences which give information about the organization of your manuscript, this will make it easier for the reader to follow your work)
2. Materials and Method (Section 2)
3. Results (Section 3)
4. Discussion (Section 4)
5. Conclusion (Section 4)
Discussion must be written completely new because this version doesn't give information about the authors' contribution to the field. In this section, authors must comment on their results and compare obtained results with those in the references if it is possible to make a comparative analysis with differences and overlapping points with the works in references.
CONCLUSION:
1. What are other evaluation metrics and classification tools in the following sentence:"In this paper, by using accuracy, F1_socre and other evaluation metrics, we demonstrate that cascaded forest models can work well in the RNA velocity prediction classification problem. And it is better than other integrated classification tools".
2. The authors must give information about the practical use of their prediction technique.
Author Response
Dear Reviewer:
Thank you for your comments concerning our manuscript entitled "Cascade Forest-based for Prediction of RNA Velocity." Those comments are all valuable and very helpful for revising and improving our paper, as well as the important guiding significance to our research. We have studied the comments carefully and made a correction that we hope meets with approval. Revised portions are marked in red on the paper. The major corrections in the paper and the responses to the reviewer's comments are as follows:
Responds to the reviewer's comments:
Point1: Clarification on why the authors set the material and method as section 4
Response1: We are very sorry for our incorrect writing. I read an article published in a journal and saw a similar typesetting, so I mistakenly thought it could be a typeset. Under your correction, I realized it was my wrong understanding, and I have reformatted the article.
Point2: In the introduction section, add sentences that give information about the organization of your manuscript. This will make it easier for the reader to follow your work.
Response2: We have made corrections according to the Reviewer's comments. We have detailed the organization of this article in the last part of the introduction so that readers can understand the work related to this article.
Point3: In this section, authors must comment on their results and compare obtained results with those in the references if it is possible to make a comparative analysis with differences and overlapping points with the works in references.
Response3: As the Reviewer suggested, We have rewritten the discussion section. In our newly authored discussion, the role of cascading forest models in this research work is revisited, focusing on comparisons with existing methods. A comprehensive comparative discussion proves that the cascade forest classifier is superior to the existing methods. The comparison shows that the cascade forest model can be applied to the relevant research of trajectory inference.
Point4: What are other evaluation metrics and classification tools in the following sentence: "In this paper, by using accuracy, F1_socre, and other evaluation metrics, we demonstrate that cascaded forest models can work well in the RNA velocity prediction classification problem. And it is better than other integrated classification tools".
Response4: We are very sorry for our incorrect writing. We have rewritten the conclusion section to make the content of the article clearer and easier to understand to avoid vague references.
Point5: The authors must give information about the practical use of their prediction technique.
Response5: We have made corrections according to the Reviewer's comments. In the rewritten conclusion, we elaborate on the practical use of their prediction technique and further explain the implications for future research work.
Special thanks to you for your good comments. We tried our best to improve the manuscript and made some changes. These changes will not influence the content and framework of the paper. And here, we did not list the changes but marked them in red in the revised paper. We appreciate the Reviewers' warm work earnestly and hope the correction will be approved. Once again, thank you very much for your comments and suggestions.
Round 2
Reviewer 2 Report
Dear authors, you answered all my concerns. Congratulations!